Internal seed dispersal by parrots: an overview of a neglected mutualism

Blanco Guillermo 1 gblanco@mncn.csic.es
Bravo Carolina 2
Pacifico Erica C. 3
Chamorro Daniel 4
Speziale Karina L. 5
Lambertucci Sergio A. 5
Hiraldo Fernando 3
Tella José L. 3
1 Department of Evolutionary Ecology, National Museum of Natural Sciences, CSIC , Madrid , Spain
2 Departament de Biologia Animal, Facultat de Biologia, Universidad de Barcelona , Barcelona , Spain
3 Department of Conservation Biology, Estación Biológica de Doñana, CSIC , Sevilla , Spain
4 Departamento de Ciencias Ambientales, Universidad de Castilla-La Mancha , Toledo , Spain
5 Ecotono Laboratory, INIBIOMA (CONICET-National University of Comahue) , Bariloche , Argentina
Wink Michael
Electronic publication date: 2016 Feb 22
Publication date: 2016
Volume: 4
Electronic Location ID: e1688
Received 2015 Dec 10; Accepted 2016 Jan 23
Copyright: ©2016 Blanco et al.
Copyright year: 2016
Copyright holder: Blanco et al.
License: This is an open access article distributed under the terms of the Creative Commons Attribution License, which permits unrestricted use, distribution, reproduction and adaptation in any medium and for any purpose provided that it is properly attributed. For attribution, the original author(s), title, publication source (PeerJ) and either DOI or URL of the article must be cited.
License URL: https://creativecommons.org/licenses/by/4.0/

Keywords: Endozoochorous seed dispersal, Fruit size, Mutualistic interactions, Psittaciformes, Vertebrate frugivores, Stomatochory

Funding: Fundación Repsol, Toyolex Veículos, Recife PICT-BID (2014) 0725 CAPES Schollarship World Parrots Trust Loro Parque Fundación CAPES-Brasil scholarship Funding was provided by Fundación Repsol, Toyolex Veículos, Recife, PE, Brazil (Instituto Arara Azul), PICT-BID (2014) 0725, CAPES Schollarship (Coordenação de Aperfeiçoamento de Pessoal de Nível Superior, Brazil). Field work on Lear’s Macaw was funded by World Parrots Trust. Lab work was funded by Loro Parque Fundación. EC Pacífico has a CAPES-Brasil scholarship. The funders had no role in study design, data collection and analysis, decision to publish, or preparation of the manuscript.

==============================
Despite the fact that parrots (Psitacifformes) are generalist apex frugivores, they have largely been considered plant antagonists and thus neglected as seed dispersers of their food plants. Internal dispersal was investigated by searching for seeds in faeces opportunistically collected at communal roosts, foraging sites and nests of eleven parrot species in different habitats and biomes in the Neotropics. Multiple intact seeds of seven plant species of five families were found in a variable proportion of faeces from four parrot species. The mean number of seeds of each plant species per dropping ranged between one and about sixty, with a maximum of almost five hundred seeds from the cacti Pilosocereus pachycladus in a single dropping of Lear’s Macaw (Anodorhynchus leari). All seeds retrieved were small (<3 mm) and corresponded to herbs and relatively large, multiple-seeded fleshy berries and infrutescences from shrubs, trees and columnar cacti, often also dispersed by stomatochory. An overview of the potential constraints driving seed dispersal suggest that, despite the obvious size difference between seeds dispersed by endozoochory and stomatochory, there is no clear difference in fruit size depending on the dispersal mode. Regardless of the enhanced or limited germination capability after gut transit, a relatively large proportion of cacti seeds frequently found in the faeces of two parrot species were viable according to the tetrazolium test and germination experiments. The conservative results of our exploratory sampling and a literature review clearly indicate that the importance of parrots as endozoochorous dispersers has been largely under-appreciated due to the lack of research systematically searching for seeds in their faeces. We encourage the evaluation of seed dispersal and other mutualistic interactions mediated by parrots before their generalized population declines contribute to the collapse of key ecosystem processes.

Introduction

How organisms mould the environment in which they live by influencing the demography and population dynamics of other organisms is a central issue in ecology (Hooper et al., 2005; Rietkerk & Van de Koppel, 2008). Factors underlying these processes are being increasingly addressed through the identification of the interacting organisms and the recognition and comprehensive understanding of the nature of their interactions (Wilson, 1992). As a consequence, a detailed natural history and the synthesis of the patterns of interaction among species are continuously merging under the consideration of overlooked ecological linkages and processes of variable complexity, and the probability of being observed in nature (Thompson, 2005; Loreau, 2010; Bascompte & Jordano, 2014).

Vertebrate frugivores have been repeatedly highlighted as key ‘mobile linkers’ with a pervasive influence in ecosystem integrity by promoting the interchange of genetic information through seed flow (Fleming & Kress, 2013; Jordano, 2014). By dispersing their food plants, frugivores can influence the composition and abundance of plant communities, thus playing a major role in ecosystem structure and functioning (Wisz et al., 2013). The importance of frugivores as plant mutualists has traditionally focused on internal dispersal (endozoochory) requiring the ingestion and subsequent defecation or regurgitation of viable seeds to be efficiently dispersed (Fleming & Kress, 2013; Jordano, 2014). Crucially, the identification of the potential dispersers is essential to fully understand dispersal mutualisms and the influence of each disperser species or group of species on the conservation of ecosystem integrity.

Among birds, effective endozoochory has been primarily attributed to fruit gulpers swallowing entire fruits and, to a lesser extent, to fruit mashers feeding on fruit pulp where small seeds can be embedded and then inadvertently swallowed (Fleming & Kress, 2013; Jordano, 2014). While the size of seeds dispersed by gulpers is constrained by gape-size, it has been argued that dispersal by fruit mashers is restricted only to minute seeds (generally smaller than 2 mm), because these frugivores discard larger seeds while biting and mandibulating fruit pulp (Wheelwright, 1985; Fleming & Kress, 2013; Jordano, 2014). Fruit mashers often also act as seed predators, leading to a wide range of interactions across an antagonism-mutualism gradient (Wheelwright & Orians, 1982; Hulme, 2002).

In particular, parrots (Psitacifformes) can alternatively feed on pulp, discarding seeds, or can actively search for large seeds that are often crushed with the bill to promote digestion, thus acting as seed predators (Janzen, 1981). However, despite the fact that most seed predators have been shown to eventually act as facultative primary dispersers (Norconk, Grafton & Conklin-Brittain, 1998; Vander Wall, Kuhn & Beck, 2005), parrots have been largely neglected as endozoochorous dispersers. Although parrots undoubtedly destroy the seeds of many plant species, they can also inadvertently or actively ingest tiny embedded in pulp and disperse them in a viable condition (Fleming et al., 1985; Oliveira, Nunes & Farias, 2012). Indeed, the Kea, Nestor notabilis has recently been shown to be the major endozoochorous disperser of alpine flora in New Zealand (Young, Kelly & Nelson, 2012). This suggests that the overlooked potential of parrots in long-distance endozoochory may have precluded the proper evaluation of bird-plant mutualistic networks, and the comprehensive understanding of evolution and coevolution of vertebrate frugivores and their food plants. If endozoochory by parrots is probed, their variable but comparatively large size, high mobility and abundance in frugivorous assemblages (Blanco et al., 2015; Marsden & Royle, 2015; Renton et al., 2016; Tella et al., 2015) can be crucial in plant life cycles and ecosystem functioning.

In this study, we evaluated whether a sample of Neotropical parrot species can defecate intact seeds of their food plants. This sampling was conceived as an exploratory study aimed to assess the potential role of parrots as endozoochorous dispersers, rather than to comprehensively evaluate internal dispersal by the sampled species or its consequences for their food plant populations, which requires specific research. Therefore, we did not systematically or seasonally search for faeces, but collected them opportunistically at communal roosts, foraging sites and nests in different habitats and biomes. We also evaluated the viability of the dispersed seeds regardless of their enhanced or limited germination capability due to the transit across the gut. Finally, we conducted an overview of the thus far largely neglected dispersal interactions between parrots and plants, in order to draw attention to their potential implications in plant-frugivore mutualistic networks and forest conservation.

Material and Methods

Fieldwork

Fresh faeces were collected at communal roosts, foraging sites and nests of eleven parrot species inhabiting different biomes, including austral and tropical dry, montane and humid forests in variable states of conservation, and urban and agro-pastoral areas, in Ecuador, Peru, Brazil, Chile and Argentina (Table 1).

Table 1 Results of the searching for seeds in parrot faeces collected in several contexts and Neotropical habitats and biomes.

For each parrot species and context, the overall proportion of faeces with seeds is shown. The proportion of faeces with seed of each plant species and the number ± SD and range of seeds per dropping are also shown. Collection data and feeding observations of parrot species for which no seed was found in the sampled faeces are also shown.

Parrot species (contexta)	Habitat, locality, date	% faeces with seeds, n	Plant species (Family)	Faeces with seeds (%)b	Mean ± SD seeds/ faeces (range)	
With seeds in sampled faeces						
Psittacara hockingi (CR)	Montane forest, Leymebamba, Perú, Dec. 2014	24.1, n = 29	Rubus sp. (Rosaceae)	6 (20.7)	3.3 ± 4.1 (1–11)	
			Maclura tinctoria (Moraceae)	1 (3.4)	1	
Thectocercus acuticaudatus (CR, FA)	Caatinga, Canudos, Brazil, Jan.–April 2015	30.2, n = 43	Pilosocereus pachycladus (Cactaceae)	13 (30.2)	25.3 ± 37.8 (1–107)	
			Tacinga inamoema (Cactaceae)	1 (2.3)	2	
Anodorhynchus leari (CR, FA)	Caatinga, Canudos, Brazil, Jan.–April 2015	49.3, n = 75	Pilosocereus pachycladus (Cactaceae)	37 (49.3)	40.9 ± 90.8 (1–481)	
			Tacinga inamoema (Cactaceae)	3 (4.0)	1.0 ± 0.0 (1)	
Anodorhynchus leari (N)	Caatinga, Canudos, Brazil, April 2015	22.2, n = 18	Pilosocereus pachycladus (Cactaceae)	2 (11.1)	8.5 ± 6.4 (4–13)	
			Cereus jamacaru (Cactaceae)	3 (16.7)	58.7 ± 80.8 (1–151)	
Myiopsitta monachus (BC)	Urban, Buenos Aires, Argentina, May 2015	8.6, n = 35	Unindentified Asteraceae	2 (5.7)	2.0 ± 0.0 (2)	
			Plantago major (Plantaginaceae)	1 (2.9)	4	
		n	Main food exploited during foraging observations			
Without seeds in sampled faeces						
Forpus coelestis (CR)	Coastal-urban, Santa Elena, Ecuador, Dec. 2014	250	Fruit pulp (Ficus sp.), nectar, bark			
Eupsittula cactorum (N)	Caatinga, Canudos, Brazil, Jan. Feb. 2015	10	Fruit pulp (Cactaceae), flowers, nectar, bark			
Amazona lilacina (CR)	Tumbesian forest, Santa Elena, Ecuador, Dec. 2014	32	Fruit pulp (mostly Spondias purpurea, Cordia lutea)			
Amazona aestiva (CR)	Caatinga, Canudos, Brazil, Jan.–April 2015	9	Fuit pulp (Cactaceae), flowers, bark			
Enicognathus ferrugineus (FA)	Urban, Bariloche, Argentina, June–Sept. 2015	42	Fruit pulp (Malus), large seeds (Prunus, Quercus), flower buds, nectar, fungi, grasses			
Enicognathus leptorhynchus (CR)	Agro-grazing, Osorno, Chile, Sept. 2015	20	Cereal grain, grasses, flower buds, fungi			
Cyanoliseus patagonus (CR)	Steppe, Junín de los Andes, Argentina, Sept. 2015	15	Cereal grain, flower buds, bark			
Notes.

a CR, communal roost; FA, foraging areas; N, nestling; BC, breeding colony.

b Note that several faeces showed the simultaneous presence of seeds of several plant species.

Faeces found beneath the trees used by communally roosting parrots were sampled early in the morning just after parrots left the roosts, which were used by single species thus precluding confusing their faeces with those of other species. Non-adjacent faeces were selected in order to avoid duplication of samples corresponding to the same individual. We also collected several faeces during observations of parrot foraging activity and during the handling of developing nestlings. Access to nests of Lear’s Macaw (Anodorhynchus leari) was authorized by the Brazilian government (permit reference SISBIO: 12763-7). Every faecal sample was collected in a paper bag, dried rapidly with a forced-air heater to prevent fungal growth and stored at room temperature until arrival at the laboratory.

The main foraging activities and the consumed part of each plant species exploited by parrots were recorded on the same dates and within the surroundings of faecal sampling sites. These observations often corresponded to the flocks attending the communal roosts and breeding areas where faeces were collected. Foraging flocks were recorded during roadside surveys at low speed, making stops to record what they were eating (Blanco et al., 2015). We recorded whether foraging parrots were feeding on pulp of ripe or unripe fruits and their mature or immature seeds, and specifically whether the consumed fruits corresponded to plants with tiny seeds that could be swallowed and pass through the gut into the faeces. The size, measured with callipers, of a sample of ripe fruits of each species consumed by parrots, as well as the number of seeds per fruit, was recorded in the field or extracted from the literature.

Laboratory work

Faeces were disaggregated on petri dishes and intact seeds were separated with the aid of binocular microscopes (20×). The seeds were immediately washed with deionized water, gently dried with laboratory blotting paper and stored in paper bags in dark conditions and at room temperature. Seeds were identified and samples of seeds of each species measured for the diameter of the smallest and largest axis to the nearest 0.1 mm with a digital calliper.

The viability of defecated seeds was determined by means of the tetrazolium test (Moore, 1985). This was aimed as an exploratory approach to assess the possibility that defecated seeds retain viability, rather than to precisely determine viability rate. Briefly, the seeds were cut and incubated in a 1% solution of 2,3,5-triphenyl tetrazolium chloride for 48 h; tetrazolium reacts with respiring radicles to produce a red stain indicating viable seeds, while non-stained white radicles indicate non-viable seeds (Moore, 1985). We further assessed the reliability of the tetrazolium test to reflect the potential germination capacity of seeds after parrot gut passage by means of a simple germination experiment; we focused on seeds of the plant more frequently recovered from the faeces. After being washed, 160 seeds of Pilosocereus pachycladus from faecal samples of A. leari were set to germinate in petri dishes (5.5 cm in diameter) over two sheets of filter paper (Filter-Lab 1300). We used eight petri dishes, with 20 seeds each. Petri dishes were incubated in a chamber at 20 °C and a photoperiod of 12 h. The petri dishes were regularly watered and sealed with parafilm to prevent them from desiccating. Germination success was scored after 60 days.

Overview of seed dispersal by parrots

We attempted to find all studies evaluating the presence of intact seeds in parrot faeces in the wild, and those experimentally testing endozoochory in captivity, by using key word searching in ISI Web of Science and Google Scholar. In addition, we surveyed dietary studies and consulted previous literature reviews on diet of parrots (e.g., Matuzak, Bezy & Brightsmith, 2008; Juniper & Parr, 2010; Renton et al., 2016) to assess the exploitation of plants with tiny seeds that could be potentially dispersed by endozoochory.

The size of seeds actually dispersed by endozoochory recorded in the present study was compared with those potentially dispersed by endozoochory and with those dispersed by stomatochory, using the data reported by Blanco et al. (2015). The size of fruits whose seeds were actually or potentially dispersed by endozoochory was also compared with those dispersed by stomatochory. This overview thus focused on preliminarily exploring the potential role of parrots as seed dispersers of their food plants by different but complementary and redundant mechanisms, and its potential implications in the evolution of fruit traits.

Results

We searched for seeds in 578 fresh faeces of 11 parrot species on different dates, and in different contexts and habitats in the Neotropics. Overall, we found 1,787 seeds of seven plant species of five families in 65 faeces from four parrot species, while the remaining seven parrot species showed no seeds in their faeces (Table 1). The proportion of faeces with seeds ranged between ∼9% and ∼49% depending on species and context (Table 1). Most faeces with seeds contained seeds from a single species (92%, n = 65), while the remaining faeces showed seeds of two species of the Cactaceae family (Table 1). The mean number of seeds of each plant species per faecal sample of each parrot species in each context ranged between 1 and 59, with a maximum of 481 seeds of P. pachycladus (Cactaceae) in a single faecal sample of A. leari (Table 1 and Fig. 1).

Figure 1 (A) Partially eaten Pilosocereus pachycladus (Cactaceae) fruit, and (B) detail of its red pulp showing multiple tiny seeds. (C) Adult Lear’s Macaw Anodorhynchus leari defecating in flight on a conspecific (probably its mate), which illustrates potential endozoochory and epizoochory. Seeds of Cereus jamacaru (D) and P. pachycladus (E) retrieved from parrot faeces. Photographs by E Pacifico (A, B), J. Marcos Rosa (C) and C Bravo (D, E).

Parrot species which revealed no seeds in their faeces were mostly foraging on multiple plant parts other than fruit, especially flower buds, nectar, bark and sprouts of native and exotic trees and shrubs, leaves, flowers, bulbs and seeds of grasses, cereal grain from agricultural and grazing areas, and wood parasitic fungi. They were also observed feeding on pulp of large-seeded fruits, both of native and exotic trees and shrubs, and predating on their seeds (Table 1). The results showing few or no seeds in faeces are clearly conservative when the sampling was conducted in seasonal periods with very low abundance of fruit/seeds or lacking fruiting plants (e.g., both Enicognathus species and Cyanoliseus patagonus sampled in late austral winter).

The mean dimensions of a sample of seeds present in the faeces is shown in Table 2 for each plant species; seeds of the same plant species found in faeces of different parrot species were pooled. These seeds usually correspond to relatively large, multiple-seeded fleshy berries and aggregates of drupes with juicy pulp from plants of variable growth forms (Table 2). Several of the plant species dispersed by endozoochory were also observed being dispersed by stomatochory (e.g., entire fruits of Rubus sp. dispersed by Psittacara hockingi, and fruits of P. pachycladus transported with the feet in flight by A. leari).

Table 2 Features of plants and fruits whose seeds were found in parrot faeces, and seed viability according to the tetrazolium test.

Plant species	Growth form	Seed size, mm (n)	Fruit type	Fruit size, mmb	No. of seedsb	Tested/viable seeds (%viable)	
Rubus sp.	Shrub	2.56 × 1.51 (6)	Berry	15.0 × 15.0	48	20/0 (0.0)	
Maclura tinctoria	Tree	2.15 × 1.32 (1)	Multiple drupe	20.0 × 12.0	50	1/0 (0.0)	
Pilosocereus pachycladus	Columnar tree-like cacti	1.89 × 1.35 (154)a	Berry	50.5 × 38.1c	3,800c	1,194/490 (41.0)	
Tacinga inamoema	Opuntiad cacti	1.98 × 1.21 (6)a	Berry	35.0 × 30.0d	Tense	4/0 (0.0)	
Cereus jamacaru	Columnar tree-like cacti	2.62 × 1.73 (20)	Berry	82.3 × 62.6f	1,400f	124/115 (92.7)	
Unindentified Asteraceae	Probably herb	2.09 × 0.81 (4)	?	?	?	4/0 (0.0)	
Plantago major	Herb	1.77 × 1.09 (2)	Capsule	5.0 × 3.5	10	4/0 (0.0)	
Notes.

a Seeds from faeces of T. acuticaudatus and A. leari.

b Approximate mean fruit size and number of seeds per fruit or infrutescence, measured in the field or extracted from the literature.

c Abud et al., 2010.

d Souza et al., 2007.

e Menezes, Taylor & Loiola, 2013.

f Abud et al., 2013.

Results of the tetrazolium test indicated that a proportion of seeds of P. pachycladus and Cereus jamacaru retrieved from faeces of two different parrot species and sampling contexts were viable (Table 2). The germination success of a sample of P. pachycladus seeds (35.6%, n = 160) was slightly less but not statistically different (Fisher’s exact test P = 0.199) than the proportion of viable seeds as assessed by the tetrazolium test (Table 2), which indicates that this test reliably reflected the potential of seeds to germinate after passing through the parrots’ gut. The seeds of the remaining species, which were found much less frequently in faeces, were inviable according to the tetrazolium test (Table 2).

Seeds dispersed by endozoochory (actual or potential) were smaller than those dispersed by stomatochory (log10 seed length, two-way ANOVA, F2,26 = 23.88, P < 0.0001, log10 seed width, F2,26 = 20.68, P < 0.0001; post-hoc tests indicated no size difference between seeds actually and potentially dispersed by endozoochory, both P > 0.05, Fig. 2). The size of fruits whose seeds were actually or potentially dispersed by endozoochory was similar for fruit length (log10 transformed, t-test, t = 0.304, P = 0.76) and slightly larger for fruit width (log10 transformed, t-test, t = 2.07, P = 0.049, n1 = 13, n2 = 15) than those dispersed by stomatochory (Fig. 3).

Figure 2 Length and width of seeds dispersed by endozoochory (blue circles, this study) and stomatochory by parrots.

Data from seeds dispersed by stomatochory (black points) and potential endozoochory (red circles) were extracted from Blanco et al. (2015).

Figure 3 Size (length and width) of fruits whose seeds were dispersed by endozoochory and stomatochory by parrots.

Data were pooled for fruits whose seeds were actually or potentially dispersed internally by parrots; data for fruits dispersed by stomatochory and potential endozoochory was extracted from Blanco et al. (2015). The boxes depict the interquartile ranges (25th–75th percentiles), the horizontal thick lines represent the medians, the black squares show the means, the whiskers extend to 1.5 times the interquartile range, and the asterisks denote the extreme cases.

Discussion

Despite the fact that parrots have traditionally been neglected as internal seed dispersers, we found seeds of several plant species in a small sample of parrot faeces collected in a variety of habitats and biomes in the Neotropics. The sampled parrot species and populations were not selected for their known frugivorous habits or local and seasonal use of fruits from particular plant species. Instead, faecal collection opportunities during the course of other studies were occasionally encountered and used to assess endozoochory, even when there were no fruiting plants on the sampling dates. As a consequence, we did not find seeds in the faeces of several of the sampled parrot species because they were not foraging on fruits during the study period, but rather on a variety of other resources. However, as trophic generalists exploiting all seasonally available feeding opportunities (e.g., Ragusa-Netto & Fecchio, 2006; Gilardi & Toft, 2012; Lee et al., 2014; Blanco et al., 2015; Renton et al., 2016), these and many other parrot species have been occasionally or frequently recorded exploiting all major neotropical plant families with tiny-seeded fruits (see diet reviews by Matuzak, Bezy & Brightsmith, 2008; Juniper & Parr, 2010; Renton et al., 2016). Therefore, the conservative results of our exploratory sampling and the literature review indicate that the importance of parrots as endozoochorous dispersers has been largely under-appreciated due to the lack of research systematically searching for seeds in their faeces.

As expected from our random sampling contexts, dates and habitats, the proportion of faeces with seeds greatly differed between plant and parrot species. Both the occurrence and number of seeds in faeces were especially high for cacti from the Caatinga, dispersed by a medium-size parakeet (T. acuticaudata) and a large macaw (A. leari); two other parrot species were recorded foraging on the same cacti species but seeds were not retrieved from their faeces, probably due to the small number of faeces analysed. The recorded figures were similar and even higher regarding the number of seeds per dropping than those reported in the literature for recognized avian frugivores (Fleming & Kress, 2013; Jordano, 2014), as also reported for the New Zealand kea (Young, Kelly & Nelson, 2012). Besides Cactaceae, we found seeds from Moraceae, Rosaceae, Asteraceae and Plantaginaceae families. Our review of the literature showed that intact seeds of other plant families have been retrieved from parrot faeces, including Mutingiaceae, Dilleniaceae, Myrtaceae, Araliaceae, Coriariaceae, Elaeocarpaceae, Ericaceae, Podocarpaceae, Polygonaceae, Rubiaceae and Lauraceae (Fleming et al., 1985; Oliveira, Nunes & Farias, 2012; Young, Kelly & Nelson, 2012; Thabethe et al., 2015). Internal dispersal of tiny seeds of these and other plant families were suspected in other studies not searching for seeds in faeces (e.g., Eitniear, Mcghee & Waddell, 1994; Norconk, Grafton & Conklin-Brittain, 1998; Contreras-González et al., 2009; Blanco et al., 2015). The variety of habits, growth forms and seed and fruit types of plants dispersed by endozoochory strengthens the key mutualist role of parrots on plant assemblages by complementing previously recorded interactions, including pollination, stomatochorous seed dispersal, seed facilitation for secondary dispersers and plant healing (Douglas, Winkel & Sherry, 2013; Blanco et al., 2015; Tella et al., 2015; Tella et al., 2016).

All seeds found in the faeces were small (<3 mm) and corresponded to herbs and relatively large, multiple-seeded fleshy berries and infrutescences with juicy pulp from columnar cacti, shrubs and trees (see also Fleming et al., 1985; Oliveira, Nunes & Farias, 2012; Young, Kelly & Nelson, 2012). Importantly, parrots are singular dispersers owing to their unique ability to simultaneously or alternatively move minute seeds from fleshy fruits by endozoochory, stomatochory and probably epizoochory (see Fig. 1C). Parrots are apparently not limited by gape size to disperse tiny seeds, although the smallest species could crush the smallest seeds, but this requires further testing. Conversely, the smallest species can be limited by body-size to disperse large seeds by stomatochory, but they can still disperse by this method seeds much larger than those dispersed by endozoochory (Boehning-Gaese, Gaese & Rabemanantsoa, 1999; Sazima, 2008; Blanco et al., 2015; Tella et al., 2015; Tella et al., 2016). Our exploratory analysis of these constraints suggests that, despite the obvious size difference between seeds dispersed by endozoochory and stomatochory, no clear differences arise for fruit size depending on the dispersal mode. This appears to be primarily due to the widespread range of size and shape of fruits dispersed by stomatochory, including those much longer than they are wide, e.g., multi-seeded pods corresponding to legumes (Fabaceae) and other large fruits (Blanco et al., 2015; Tella et al., 2015; Tella et al., 2016). Seed dispersal mutualisms mediated by parrots can thus have multiple potential implications for the understanding of bird-fruit interactions, especially because only internal seed dispersal constrained by gape size has been generally considered as an evolutionary force selecting for avian-dispersed seed size (Wheelwright, 1985; Fleming & Kress, 2013; Galetti et al., 2013; Jordano, 2014).

A large proportion of the cacti seeds frequently found in the faeces of two parrot species were viable according to the tetrazolium test. The maximum germination success in laboratory conditions of seeds extracted from mature fruits of C. jamacaru (94.0%: Meiado et al., 2010, 89.0%: Abud et al., 2013, both at 25 oC and 12 h photoperiod) was similar to the proportion of viable seeds of the same cacti retrieved from A. leari faeces (92.7%). Germination success recorded by Abud et al. (2010) for P. pachycladus seeds in the same conditions was, however, higher than the proportion of viable seeds and germination success of seeds from parrot faeces recorded in this study. Sample size of those species for which seeds were inviable (≤20 seeds in all cases) was insufficient to adequately determine this trait, given the variable natural viability of seeds (Long et al., 2015) and the low germinability of several of the recorded species (e.g., <35% in Tacinga inamoema, Nascimento et al., 2015). Seed viability, especially of those seeds from the plant species less frequently found in faeces, could also be affected by the seed drying conditions carried out in the field, and the subsequent storage after analysis, or they may actually be affected by passage through the parrots’ gut. Alternatively, these seeds could correspond to immature fruits often exploited by parrots (Norconk, Grafton & Conklin-Brittain, 1998; Blanco et al., 2015). In any case, our exploratory experiments and findings add to those of previous studies suggesting that parrots can be endozoochorous dispersers enhancing or limiting seed germinability to variable extents depending on plant and parrot species (Fleming et al., 1985; Oliveira, Nunes & Farias, 2012; Thabethe et al., 2015), as stated for recognized avian seed dispersers (Traveset, Robertson & Rodríguez-Pérez, 2007).

In conclusion, despite the fact that parrots constitute an evolutionarily ancient, highly diversified and widely distributed group of generalist apex frugivores (Toft & Wright, 2015), they have been largely overlooked as seed dispersers of their food plants, and thus excluded from animal–plant interaction networks (Fleming & Kress, 2013; Bascompte & Jordano, 2014). This exclusion has likely been promoted by the relatively large size, canopy use and high mobility of parrots, hindering detailed observations of stomatochory (Blanco et al., 2015; Tella et al., 2015) and, especially due to the difficulty of mist-netting them to collect faeces to evaluate endozoochory. Importantly, this knowledge gap implies a markedly biased view of frugivore-plant mutualistic interactions towards fruit gulpers, despite the fact that parrots constitute rich species guilds showing a greater range of size, morphology and foraging behaviours, and accounting for a higher density and biomass than other recognized frugivores in many tropical and temperate ecosystems (Blanco et al., 2015; Marsden & Royle, 2015; Renton et al., 2016; Toft & Wright, 2015). This supports the emerging view that many species traditionally regarded only as seed predators can also act as pervasive seed dispersers owing to their comparatively high abundance (Heleno et al., 2011; Orłowski et al., 2016). Worryingly, in addition to the loss of frugivore–plant interactions (so far mostly focused on fruit gulpers) due to forest destruction and fragmentation (Markl et al., 2012; Sebastián-González et al., 2015), the intensive persecution and capture of parrots for the pet trade may be decimating populations of once common species (Tella & Hiraldo, 2014; Annorbah, Collar & Marsden, 2016; Toft & Wright, 2015), thus disrupting largely unknown mutualistic interactions between parrots and their food plants. We encourage an comprehensive evaluation of seed dispersal and other mutualistic interactions mediated by parrots before their generalized population declines contribute to the collapse of key ecosystem processes.

We thank G Manzini, P Plaza, O Mastrantuoni, T Filadelfo, D Alves, M Cardoso, J. Carlos Nogueira, M. F. Lacerda da Silva, F Riera, T Valença, AO Bermúdez and Fundação Biodiversitas (Canudos Biological Station) for field-work assistance. J. Marcos Rosa kindly cedes his Lear’s Macaw photos. We acknowledge the efforts of S Tollington and two anonymous reviewers in improving the paper’s content.

Additional Information and Declarations

Competing Interests

Author Contributions

Animal Ethics

Field Study Permissions

Data Availability

The authors declare there are no competing interests.

Guillermo Blanco conceived and designed the experiments, performed the experiments, analyzed the data, contributed reagents/materials/analysis tools, wrote the paper, prepared figures and/or tables, reviewed drafts of the paper.

Carolina Bravo, Erica C. Pacifico, Daniel Chamorro, Karina L. Speziale and Sergio A. Lambertucci performed the experiments, contributed reagents/materials/analysis tools, reviewed drafts of the paper.

Fernando Hiraldo and José L. Tella conceived and designed the experiments, performed the experiments, contributed reagents/materials/analysis tools, reviewed drafts of the paper.

The following information was supplied relating to ethical approvals (i.e., approving body and any reference numbers):

Our study involved observations with binoculars and collection of parrot faeces. The animals were not manipulated for the collection of faeces but for regular monitoring in the course of other projects.

The following information was supplied relating to field study approvals (i.e., approving body and any reference numbers):

Brazilian government permit: SISBIO 12763-7.

The following information was supplied regarding data availability:

All the raw data in the article is presented in the tables. Some of the data used in several analyses and figures are from another open-access article cited in the text (Blanco et al., 2015).

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
