# Peer review of "Internal seed dispersal by parrots: an overview of a neglected mutualism"

_PeerJ, doi:10.7717/peerj.1688_

## Round 0.1 · original submission · Minor Revisions

Dear authors

thank you for submitting your ms to our journal. We have received 3 reviews. We would encourage you to revise your ms according to the suggestions of the reviewers.

Kind regards

Michael Wink
Editor

Reviewer 1 ·

Basic reporting

No Comments

Experimental design

no comments

Validity of the findings

No Comments

Additional comments

Line 26: for most Psittaciformes species the asseveration of “high densities… in many… ecosystems” is no longer valid. It is valid for some of them but not for “many”. Thus, I would not add this comment as the first sentence of the abstract. It’ll generate a false impression among non-specialist readers.

Line 28: I would discourage the use of terms like “Endozoochorous” and other technical words in the abstract. The abstract should also aim to non-specialist readers. Besides, these words are already mentioned in the Keywords, what is the important thing regarding search machines.

Abstract: I would suggest not giving values (=numbers) in the abstract. Instead, I would suggest expressing the main results with words easily understandable to all readers. Then, your work will have more chances to be read.

Lines 102-103, 109: I suggest using “exploratory” but not “preliminary”. Idem in other paragraphs.

Introduction: Tella et al. (2015) “Parrots as overlooked seed dispersers” should be mentioned in this section.

Table 1, caption: it needs to be rewritten. It starts with “Presence of intact seeds”, when the authors also give data on non-seed items. Thus, I would suggest starting with a more general description. Also, what the authors exactly mean by “Foraging observations”. Were these items found in the feces or were the birds observed feeding on these items? The table should be fully understandable without reading the text.

Table 1: I think it’ll be better “Unindentified Asteraceae”, without brackets.

Fig. 1, caption, “note the red colour of the faecal material corresponding to feeding on P. pachycladus fruits”: honestly, I cannot see this. I think one cannot tell looking only at the picture in its present resolution. Without providing additional proof, this sentence is pure speculation.

Lines 158-164: I do not see the point of this paragraph. It is common practice to search the literature at the time of writing a MS. The authors do not carry any literature statistical analyses. They use it in the usual way for the Intro and Discussion.

Reviewer 2 ·

Basic reporting

This manuscript describes a study of the extent to which parrots act as seed dispersers, by examining the number of seeds present in faeces collected opportunistically across a number of neotropical parrot species. The manuscript presents these data alongside a review of the literature on the role of parrots as seed dispersers. Statistical tests identify differences in observed proportion of seeds and some morphometrics taken from them. For faeces samples collected from one parrot species the authors test the viability of the seeds using two different methods. The authors then relate the results to literature reviewed. The analyses appear to be appropriate given the data and the conclusions drawn do not misinterpret the results.

Experimental design

The study appears to be opportunistic in terms of collection of faeces samples. The choice of the parrot species for the seed viability tests (lines 150-156) seems to be based on the fact that this species yielded the largest sample size compared to the other species included in the study; this is satisfactory but a clear justification needs to be included. The methods regarding the literature review do not have any details on what key words were used, or numbers of studies identified. I would recommend that the authors include this information in order that others may replicate their work in the future.

Validity of the findings

The findings appear to be valid, with the caveat that the sampling strategy for the collection of faeces samples was largely opportunistic.

Additional comments

Overall, this study adds to the existing literature on parrot seed dispersal and the accompanying review of the existing literature adds value. A number of minor suggested edits are below;
Line 59: delete ‘overwhelming’
Line 60: ‘continuously feeding back and merging’ – this is confusing. Rephrase. The sentence is also too long.
Line 66: how can demography be enhanced? – rephrase.
Line 72: ‘the major among’ – does not make sense. Correct.
Line 89: ‘This [what?] is striking’.
Line 97: ‘could be probed widespread’ – doesn’t make sense. Rephrase.
Line 99: ‘could be presumed crucial’ - doesn’t make sense. Rephrase.
Line 110: ‘in order to call attention on’ – change to read ‘in order to draw attention to’
Line 199: Why not spell out what proportion exactly?
Line 231: delete ‘clearly’
Line 284: change to ‘Sample size of those species for which seeds were viable’
Line 299: what are genetic linkers? – Rephrase.
Line 309: Delete ‘showing’
Line 310 delete ‘assumed’ and replace with ‘regarded’

Line 317: Delete ‘urgent’.
Fig 2: Legend needs to spell out what the red and blue circles relate to.

·

Basic reporting

I have annotated the attached pdf document with my suggested edits to the manuscript text.

Experimental design

This paper highlights an important knowledge gap with an opportunistic experimental approach which the authors are transparent about. Whilst not being a comprehensive or systematic experiment the findings are important.

Validity of the findings

I agree with the authors that importance of parrots as endozoochorous dispersers
has been under-appreciated and under studied. Therefore this is an important demonstration and call for further research.

Additional comments

The aims and scope of PeerJ state that it does not accept literature reviews, whilst this paper is not 100% review it does contain a significant review proportion. Whether this is too much is a decision for the Editor. What I would at least recommend is that the title is amended; the word 'overview' in the title suggests a review.

---

## Round 0.2 · accepted · Accept

Dear authors
thank you for revising your manuscript. Therefore, we can accept your ms by now.
Regards

Michael Wink